# Identifying Phase Transition Thresholds of Permuted Linear Regression via Message Passing

## Abstract

This paper considers the permuted linear regression, i.e., $\mathbf{Y} = \mathbf{\Pi}^{\natural}\mathbf{X}\mathbf{B}^{\natural} + \mathbf{W}$, where $\mathbf{Y} \in \mathbb{R}^{n \times m}, \mathbf{\Pi}^{\natural} \in \mathbb{R}^{n \times n}, \mathbf{X} \in \mathbb{R}^{n \times p}, \mathbf{B}^{\natural} \in \mathbb{R}^{p \times m}$, and $\mathbf{W} \in \mathbb{R}^{n \times m}$ represent the observations, missing (or incomplete) information about ordering, sensing matrix, signal of interests, and additive sensing noise, respectively. As is shown in the previous work, there exists phase transition phenomena in terms of the *signal-to-noise ratio* (snr), number of permuted rows, etc. While all existing works only concern the convergence rates without specifying the associate constants in front of them, we give a precise identification of the phase transition thresholds via the message passing algorithm. Depending on whether the signal $\mathbf{B}^{\natural}$ is known or not, we separately identify the corresponding critical points around the phase transition regimes. Moreover, we provide numerical experiments and show the empirical phase transition points are well aligned with theoretical predictions.

## 1 Introduction

This paper considers the permuted linear regression

$$\mathbf{Y} = \mathbf{\Pi}^{\natural}\mathbf{X}\mathbf{B}^{\natural} + \sigma\mathbf{W},$$

where $\mathbf{Y} \in \mathbb{R}^{m \times n}$ denotes the sensing result, $\mathbf{\Pi}^{\natural} \in \mathbb{R}^{n \times n}$ represents the permutation matrix, $\mathbf{X} \in \mathbb{R}^{n \times p}$ is the sensing matrix, $\mathbf{B}^{\natural}$ is the signal of interests, $\mathbf{W}$ denotes the additive noise, and $\sigma$ is the noise variance. The research on this problem dates back at least to 1970s under the name 'broken sample problem' (Goel, 1975; Bai & Hsing, 2005; DeGroot et al., 1971; DeGroot & Goel, 1976; 1980). In recent years, we have witnessed a revival of this problem due to its board spectrum of applications in privacy protection, data integration, etc (Pananjady et al., 2018; Unnikrishnan et al., 2015; Slawski et al., 2020; Slawski & Ben-David, 2019; Pananjady et al., 2017; Zhang et al., 2022; Zhang & Li, 2020).

Associated with this problem comes a phase transition phenomenon: the error rate for the permutation recovery suddenly drops to zero once some parameters reach certain thresholds. Despite previous work such as Slawski et al. (2020); Slawski & Ben-David (2019); Pananjady et al. (2017); Zhang et al. (2022); Zhang & Li (2020) can all explain this phenomenon, the precise positions of the phase transition thresholds are never studied but rather their statistical order. In this work, we would like to leverage *message passing* (MP) algorithm to identify their precise location. As a byproduct, we also come up with an algorithm to partially recover the permutation matrix.

**Related work.** The line of research starts with the literature in permuted linear regression. Among all the works mentioned above, the most related works include Slawski et al. (2020); Slawski & Ben-David (2019); Pananjady et al. (2017); Zhang et al. (2022); Zhang & Li (2020), in which almost the same settings as ours are used. Pananjady et al. (2018); Slawski & Ben-David (2019) consider the single observation model ($m = 1$) and proved the snr for the correct permutation recovery will be $\mathbb{O}_{\mathrm{P}}(n^c)$, where $c > 0$ is some positive constant. Later, Slawski et al. (2020); Zhang et al. (2022); Zhang & Li (2020) investigate the multiple observations model ($m > 1$) and suggest the snr requirement can be significantly decreased, to put it more specifically, from $\mathbb{O}_{\mathrm{P}}(n^c)$ to $\mathbb{O}_{\mathrm{P}}(n^{c/m})$. In particular, Zhang & Li (2020) develop an estimator which we will analyze as part of our contributions. Although they obtained the correct convergence rate to restore the correspondence,

which are minimax-optimal in certain regimes, their results fail to specify the leading coefficients, or equivalently, the precise location of the phase transition threshold. Moreover, their analysis does not consider the intertwined influence among the parameters $n$, $p$, $m$, etc. One example would be the impact of the $p/n$ ratio on the maximum allowed number of permuted rows, which has not been studied before this work.

Another line of research comes from the field of statistical physics, which begins with Mézard & Parisi (1986; 1985). Using the replica method, they study the *linear assignment problem* (LAP), i.e., $\min_\Pi \sum_{i,j} \Pi_{ij} E_{ij}$ where $\Pi$ denotes a permutation matrix and $E_{ij}$ is i.i.d random variable uniformly distributed within the regime $[0, 1]$. Martin et al. (2005) then generalize the LAP to multi-index matching and presented a investigation based on MP algorithm. And Caracciolo et al. (2017); Malatesta et al. (2019) extend the distribution of $E_{ij}$ to a broader class. However, all the above works exhibit no phase transition. In Chertkov et al. (2010), this method is extended to the particle tracking problem, where a phase transition phenomenon is first observed. Later, Semerjian et al. (2020) modify it to fit the graph matching problem, which paves way for our work in studying the permuted linear regression.

Our technical **contributions** are summarized as follows

- We propose the first framework that can identify the precise location of phase transition thresholds associated with permuted linear regression. In the oracle case where $\mathbf{B}^\natural$ is known, our scheme is able to determine the phase transition snr. In the non-oracle case where $\mathbf{B}^\natural$ is not given, our scheme can further predict the maximum allowed permuted rows and uncover its dependence on the ratio $n/p$.

- We generalize the full permutation estimator and first obtain a partial permutation estimator. Consider the example where the correspondence for a single index is desired. By removing all function nodes except that corresponds to that index, we exploit the MP algorithm and design an algorithm that converge in one step. Moreover, we show its performance almost match the estimator for the full permutation recovery.

In addition, we would like to briefly mention the technical challenges. Compared with the previous works (Mezard & Montanari, 2009; Talagrand, 2010; Linusson & Wästlund, 2004; Mézard & Parisi, 1987; 1986; Parisi & Ratiéville, 2002; Semerjian et al., 2020), where the edge weights are relatively simple, our edge weights usually involve high-order interactions across Gaussian random variables and are densely correlated. To tackle this issue, our proposed approximation method to compute the phase transition thresholds consists of three parts: $(i)$ perform Taylor expansion; $(ii)$ modify the leave-one-out technique; and $(iii)$ size correction scheme. A detailed explanation can be found in Section 5. Hopefully, it will serve independent technical interests for researchers in the machine learning community.

**Notations.** We use $a \xrightarrow{\text{a.s.}} b$ to suggest $a$ converges almost surely to $b$. We denote $f(n) \cong g(n)$ when $\lim_{n\to\infty} f(n)/g(n) = 1$. We denote $f(n) = \mathbb{O}_\mathsf{P}(g(n))$ if the sequence $f(n)/g(n)$ is bounded in probability; while we denote $f(n) = o_\mathsf{P}(g(n))$ if the sequence $f(n)/g(n)$ converges to zero in probability. The inner product between two vectors (resp. matrices) are denoted as $\langle \cdot, \cdot \rangle$. In addition, for two distributions $d_1$ and $d_2$, we write $d_1 \cong d_2$ if they are the equal up to some normalization. Moreover, we define $\mathcal{P}_n$ as the set of all possible permutation matrices, i.e., $\mathcal{P}_n \triangleq \{\mathbf{\Pi} \in \{0, 1\}^{n\times n}, \sum_i \mathbf{\Pi}_{ij} = 1, \sum_j \mathbf{\Pi}_{ij} = 1\}$; and associate each permutation matrix $\mathbf{\Pi} \in \mathcal{P}_n$ with a mapping $\pi$ of $\{1, 2, \ldots, n\}$, where $\pi(i)$ denotes the correspondence of index $i$ permuted by $\mathbf{\Pi}$, $1 \leq i \leq n$. The *signal-to-noise-ratio* (snr) is written as $\|\mathbf{B}^\natural\|_\mathrm{F}^2/(m \cdot \sigma^2)$, where $\|\cdot\|_\mathrm{F}$ is the Frobenius norm and $\sigma^2$ denotes the variance of the sensing noise.

## 2 PROBLEM SETTING

In this paper, we consider the linear regression with permuted labels reading as

$$\mathbf{Y} = \mathbf{\Pi}^\natural \mathbf{X} \mathbf{B}^\natural + \sigma \mathbf{W},$$

where $\mathbf{Y} \in \mathbb{R}^{n\times m}$ represents the sensing result, $\mathbf{\Pi}^\natural \in \mathcal{P}_n$ denotes a permutation matrix awaiting to be reconstructed, $\mathbf{X} \in \mathbb{R}^{n\times p}$ is the sensing matrix with each entry $X_{ij}$ following the i.i.d standard

normal distribution, $\mathbf{B}^\natural \in \mathbb{R}^{p \times m}$ is the signal of interests, and $\mathbf{W} \in \mathbb{R}^{n \times m}$ represents the additive sensing noise and its entries $\mathbf{W}_{ij}$ are i.i.d standard normal random variables. In addition, we denote $h$ as the Hamming distance between the identity matrix and the permutation matrix $\mathbf{\Pi}^\natural$, i.e., $h \triangleq \sum_i \mathbb{1}(\pi^\natural(i) \neq i)$.

The goal is to reconstruct the permutation matrix $\mathbf{\Pi}^\natural$ from the pair $(\mathbf{Y}, \mathbf{X})$. As is well known in the previous works (Zhang & Li, 2020; Pananjady et al., 2018; Zhang et al., 2022), there exists a phase transition phenomenon inherent in this problem. However, all these work only present the statistical order without specifying the constants. In this work, we would like to identify the precise position of the phase transition points in the large-system limit, i.e., $n, m, p$, and $h$ all approach to infinity with $m/n \to \tau_m, p/n \to \tau_p$ and $h/n \to \tau_h$. [1]

Inspired by the Mezard & Montanari (2009); Semerjian et al. (2020); Chertkov et al. (2010), we borrow the tools from the statistical physics to identify the precise location of the phase transition threshold. In the following context, we separately study the phase transition phenomenon in $(i)$ the oracle case where $\mathbf{B}^\natural$ is given as a prior and $(ii)$ the non-oracle case where $\mathbf{B}^\natural$ is unknown.

## 3 BACKGROUND KNOWLEDGE ON GRAPHICAL MODELS

To begin with, we briefly review the *linear assignment problem* (LAP), which is defined as

$$\widehat{\mathbf{\Pi}} = \mathrm{argmin}_{\mathbf{\Pi} \in \mathcal{P}_n} \langle \mathbf{\Pi}, -\mathbf{E} \rangle, \tag{1}$$

where $\mathbf{E} \in \mathbb{R}^{n \times n}$ is a fixed matrix and $\mathcal{P}_n$ denotes the set of all possible permutation matrices. The following context investigates the behavior of $\widehat{\mathbf{\Pi}}$ with the *message-passing* (MP) algorithm. First, we follow the approach in Semerjian et al. (2020); Mezard & Montanari (2009) and introduce a probability measure over the permutation matrix $\mathbf{\Pi}$, which reads as

$$\mu(\mathbf{\Pi}) = Z^{-1} \prod_i \mathbb{1}\big(1 - \sum_j \Pi_{ij}\big) \prod_j \mathbb{1}\big(1 - \sum_i \Pi_{ij}\big) e^{-\beta \sum_{i,j} \Pi_{ij} E_{ij}}, \tag{2}$$

where $\mathbb{1}(\cdot)$ is the indicator function, $Z$ is the normalization constant of the probability measure $\mu(\mathbf{\Pi})$, and $\beta > 0$ is an auxiliary parameter. We can verify that the solution to the ML estimator maximizes the probability measure $\mu(\mathbf{\Pi})$, which means we can study the properties of $\widehat{\mathbf{\Pi}}$ via the configuration $\mathrm{argmax}_{\mathbf{\Pi}} \mu(\mathbf{\Pi})$. In addition, we notice the probability measure $\mu(\mathbf{\Pi})$ concentrates on $\widehat{\mathbf{\Pi}}$ when letting $\beta \to \infty$. Then we identify the phase transition thresholds by studying the marginals of $\mu(\mathbf{\Pi})$.

### 3.1 CONSTRUCTION OF GRAPHICAL MODEL

To start with, we construct the factor graph associated with the probability measure in (2). Adopting the same strategy as in Chapter 16 in Mezard & Montanari (2009), we $(i)$ associate each variable $\Pi_{ij}$ a variable node $v_{ij}$; $(ii)$ connect the variable node $v_{ij}$ a function node representing the term $e^{-\beta \Pi_{ij} E_{ij}}$; and $(iii)$ associate each constraint $\sum_i \Pi_{ij} = 1$ one function node and similarly for the constraint $\sum_j \Pi_{ij} = 1$. A graphical representation is put in Figure 1.

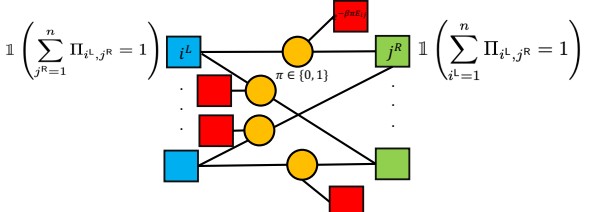

Figure 1: Illustration of the constructed graphical model. The circle icons represents the variable node; while the square icons represent the function node: the blue square icon represents the constraints for the rows of $\mathbf{\Pi}$, the green square icon represents the constraints for the columns of $\mathbf{\Pi}$, and the red square icon denotes the function $e^{-\beta \pi E_{ij}}$.

Then we briefly review the MP algorithm. Informally speaking, MP is local algorithm to compute the marginal probabilities over the graphical model. In each iteration, the variable node $v$ transmits the message to its incident function node $f$ by multiplying all incoming messages except that along the edge $(v, f)$. And the function

---

[1] Although our analysis concerns the large-system limit, numerical results matches our predicted results to a good extent even when $n, m, p$, and $h$ are a few hundreds.

node $f$ transmits the message to its incident variable node $v$ by computing the weighted summary of all incoming messages except that along the edge $(f, v)$. For a detailed introduction to MP, we refer readers to Kschischang et al. (2001); Mezard & Montanari (2009).

MP is able to obtain the exact marginals (Mezard & Montanari, 2009) for the tree-like graphical models. While for graphs with a lot of short loops, which happens to be our case, this claim may become invalid. However, past works all suggest that MP can still obtain meaningful results when applying to LAP (Mezard & Montanari, 2009; Semerjian et al., 2020).

## 3.2 MESSAGE PASSING (MP) ALGORITHM

Next, we turn to the permutation recovery via MP. The following derivation follows the standard procedure, which can be found in the previous works (Semerjian et al., 2020; Mezard & Montanari, 2009). Denote the message flow from the node $i^{\mathsf{L}}$ to the variable node $(i^{\mathsf{L}}, j^{\mathsf{R}})$ as $\widehat{m}_{i^{\mathsf{L}} \to (i^{\mathsf{L}}, j^{\mathsf{R}})}(\cdot)$ and that from the edge $(i^{\mathsf{L}}, j^{\mathsf{R}})$ to node $i^{\mathsf{L}}$ as $m_{(i^{\mathsf{L}}, j^{\mathsf{R}}) \to i^{\mathsf{L}}}(\cdot)$. Similarly, we define $\widehat{m}_{j^{\mathsf{R}} \to (i^{\mathsf{L}}, j^{\mathsf{R}})}(\cdot)$ and $\widehat{m}_{(i^{\mathsf{L}}, j^{\mathsf{R}}) \to j^{\mathsf{R}}}(\cdot)$ as the message flow transmitted between the functional node $j^{\mathsf{R}}$ and the variable node $(i^{\mathsf{L}}, j^{\mathsf{R}})$. Here the superscripts $\mathsf{L}$ and $\mathsf{R}$ are used to indicate the positions of the node (left and right). Roughly speaking, these transmitted messages can be viewed as (unnormalized) conditional probability $\mathbb{P}(\Pi_{i,j} = \{0, 1\}|(\cdot))$ with the joint pdf being defined in (2). And the message transmission process is to iteratively compute these conditional probabilities. For a more detailed introduction of the MP algorithm, we refer to Mezard & Montanari (2009); MacKay et al. (2003).

First, we consider the message flows transmitted between the functional node $i^{\mathsf{L}}$ and the variable node $(i^{\mathsf{L}}, j^{\mathsf{R}})$, which are written as

$$m_{(i^{\mathsf{L}}, j^{\mathsf{R}}) \to i^{\mathsf{L}}}(\pi) \cong \widehat{m}_{j^{\mathsf{R}} \to (i^{\mathsf{L}}, j^{\mathsf{R}})}(\pi) e^{-\beta \pi E_{i^{\mathsf{L}}, j^{\mathsf{R}}}};$$

$$\widehat{m}_{i^{\mathsf{L}} \to (i^{\mathsf{L}}, j^{\mathsf{R}})}(\pi) \cong \sum_{\pi_{i^{\mathsf{L}}, k^{\mathsf{R}}}} \prod_{k^{\mathsf{R}} \neq j^{\mathsf{R}}} \widehat{m}_{k^{\mathsf{R}} \to (i^{\mathsf{L}}, k^{\mathsf{R}})}(\pi_{i^{\mathsf{L}}, k^{\mathsf{R}}}) e^{-\beta \pi_{i^{\mathsf{L}}, k^{\mathsf{R}}} E_{i^{\mathsf{L}}, k^{\mathsf{R}}}} \mathbb{1}(\pi + \sum_{k} \pi_{i^{\mathsf{L}}, k^{\mathsf{R}}} = 1), \quad (3)$$

where $\pi \in \{0, 1\}$ is a binary data. Similarly, we can write the message flows between the functional node $j^{\mathsf{R}}$ and the variable node, which are denoted as $(i^{\mathsf{L}}, j^{\mathsf{R}})$ as $m_{(i^{\mathsf{L}}, j^{\mathsf{R}}) \to j^{\mathsf{R}}}(\pi)$ and $\widehat{m}_{j^{\mathsf{R}} \to (i^{\mathsf{L}}, j^{\mathsf{R}})}(\pi)$, respectively. With the parametrization method, we define

$$h_{i^{\mathsf{L}} \to (i^{\mathsf{L}}, j^{\mathsf{R}})} \triangleq \frac{1}{\beta} \log \frac{\widehat{m}_{i^{\mathsf{L}} \to (i^{\mathsf{L}}, j^{\mathsf{R}})}(1)}{\widehat{m}_{i^{\mathsf{L}} \to (i^{\mathsf{L}}, j^{\mathsf{R}})}(0)}; \quad h_{j^{\mathsf{R}} \to (i^{\mathsf{L}}, j^{\mathsf{R}})} \triangleq \frac{1}{\beta} \log \frac{\widehat{m}_{j^{\mathsf{R}} \to (i^{\mathsf{L}}, j^{\mathsf{R}})}(1)}{\widehat{m}_{j^{\mathsf{R}} \to (i^{\mathsf{L}}, j^{\mathsf{R}})}(0)}.$$

Define $\zeta$ as $h_{i^{\mathsf{L}} \to (i^{\mathsf{L}}, j^{\mathsf{R}})} + h_{j^{\mathsf{R}} \to (i^{\mathsf{L}}, j^{\mathsf{R}})} - E_{i^{\mathsf{L}}, j^{\mathsf{R}}}$, we select the edge $(i^{\mathsf{L}}, j^{\mathsf{R}})$ according to the probability $m_{(i^{\mathsf{L}}, j^{\mathsf{R}})}(\pi) \triangleq \frac{\exp(\beta \pi \zeta_{i^{\mathsf{L}}, j^{\mathsf{R}}})}{1 + \exp(\beta \zeta_{i^{\mathsf{L}}, j^{\mathsf{R}}})}$, $\pi \in \{0, 1\}$. Provided $m_{(i^{\mathsf{L}}, j^{\mathsf{R}})}(1) > m_{(i^{\mathsf{L}}, j^{\mathsf{R}})}(0)$, or equivalently,

$$\zeta_{i^{\mathsf{L}}, j^{\mathsf{R}}} > 0, \quad (4)$$

we pick $\widehat{\pi}(i^{\mathsf{L}}) = j^{\mathsf{R}}$; otherwise, we have $\widehat{\pi}(i^{\mathsf{L}}) \neq j^{\mathsf{R}}$. Due to the fact that $\mu(\mathbf{\Pi})$ concentrates on $\widehat{\mathbf{\Pi}}$ when $\beta$ is sufficiently large, we can thus simply the MP update equation as

$$h_{i^{\mathsf{L}} \to (i^{\mathsf{L}}, j^{\mathsf{R}})} = \min_{k^{\mathsf{R}} \neq j^{\mathsf{R}}} E_{i^{\mathsf{L}}, k^{\mathsf{R}}} - h_{k^{\mathsf{R}} \to (i^{\mathsf{L}}, k^{\mathsf{R}})}; \quad h_{j^{\mathsf{R}} \to (i^{\mathsf{L}}, j^{\mathsf{R}})} = \min_{k^{\mathsf{L}} \neq i^{\mathsf{L}}} E_{k^{\mathsf{L}}, j^{\mathsf{R}}} - h_{k^{\mathsf{L}} \to (k^{\mathsf{L}}, j^{\mathsf{R}})}, \quad (5)$$

which is obtained by letting $\beta \to \infty$.

## 4 ANALYSIS OF ORACLE CASE

As a warm-up example, we first consider the oracle scenario, where $\mathbf{B}^{\natural}$ is given a prior. To reconstruct the permutation matrix $\mathbf{\Pi}^{\natural}$, we adopt the *maximum-likelihood* (ML) estimator reading as

$$\widehat{\mathbf{\Pi}}^{\mathsf{oracle}} = \mathrm{argmax}_{\mathbf{\Pi}} \left\langle \mathbf{\Pi}, \mathbf{Y}\mathbf{B}^{\natural\top}\mathbf{X}^{\top} \right\rangle, \ \text{s.t.} \ \sum_{i} \mathbf{\Pi}_{ij} = 1, \sum_{j} \mathbf{\Pi}_{ij} = 1, \mathbf{\Pi} \in \{0, 1\}^{n \times n}. \quad (6)$$

Denote the variable $\mathbf{E}_{ij}^{\mathsf{oracle}}$ as $-\mathbf{X}_{\pi^{\natural}(i)}^{\top}\mathbf{B}^{\natural}\mathbf{B}^{\natural\top}\mathbf{X}_{j} + \sigma \mathbf{W}_{i}^{\top}\mathbf{B}^{\natural\top}\mathbf{X}_{j}, (1 \leq i, j \leq n)$, we can transform the objective function in (6) as the canonical form of LAP, i.e., $-\sum_{i,j} \mathbf{\Pi}_{ij} \mathbf{E}_{ij}^{\mathsf{oracle}}$.

## 4.1 Identifying the Phase Transition Threshold

This subsection studies the phase transition phenomenon inherent in the MP update equation (5). Following the same strategy as in Semerjian et al. (2020), we divide all edges $(i^\mathsf{L}, j^\mathsf{R})$ into two categories based on whether the edge $(i^\mathsf{L}, j^\mathsf{R})$ corresponds to the ground-truth permutation matrix $\mathbf{\Pi}^\natural$ or not. Within each category, we assume the edges's weights and the message flows along them are independent identically distributed. For the edge $(i^\mathsf{L}, \pi^\natural(i^\mathsf{L}))$ corresponding to the ground-truth correspondence, we represent its weight as a random variable called $\Omega$ and the associated message flow as a random variable called $H$ (both $h_{i^\mathsf{L} \to (i^\mathsf{L}, j^\mathsf{R})}$ and $h_{j^\mathsf{R} \to (i^\mathsf{L}, j^\mathsf{R})}$). Similarly, we define random variables $\widehat{\Omega}$ and $\widehat{H}$ for other edges. Then we can rewrite (5) as

$$\widehat{H}^{(t+1)} = \min\left(\Omega - H^{(t)}, H^{'(t)}\right), \qquad H^{(t+1)} = \min_{1 \le i \le n-1} \widehat{\Omega}_i - \widehat{H}_i^{(t)}, \tag{7}$$

where $(\cdot)^{(t)}$ denotes the update in the $t$th iteration, $H'$ is an independent copy of $H$, and $\{H_i^{(t)}\}_{1 \le i \le n-1}$ and $\{\widehat{\Omega}_i\}_{1 \le i \le n-1}$ denote the i.i.d. copies of random variables $H_{(\cdot)}^{(t)}$ and $\widehat{\Omega}_{(\cdot)}$.

Then we turn to computing the critical point where the permutation matrix can be perfectly reconstructed. According to (4), this means the event $H + H' > \Omega$ holds with probability one. Conditional on this event, we can simplify (7) to be

$$H^{(t+1)} = \min_{1 \le i \le n-1} H_i^{(t)} + \Xi_i, \tag{8}$$

where the random variable $\Xi$ is defined as the difference between $\widehat{\Omega}$ and $\Omega$, i.e., $\Xi \triangleq \widehat{\Omega} - \Omega$; and $\{H_i^{(t)}\}_{1 \le i \le n-1}$ and $\{\Xi_i\}_{1 \le i \le n-1}$ denote the i.i.d. copies of random variables $H_{(\cdot)}^{(t)}$ and $\Xi_{(\cdot)}$.

This equation can be viewed as the analogous version of the *density evolution* and *state evolution*, which are used to analyze the convergence of the message passing and approximate message passing algorithm, respectively (Chung, 2000; Richardson & Urbanke, 2001; 2008; Maleki, 2010; Donoho et al., 2009; Bayati & Montanari, 2011; Rangan, 2011). Adopting the same viewpoint of Semerjian et al. (2020), we treat (8) as a *branching random walk* (BRW) process, which satisfies

**Theorem 1** ((Biggins, 1977; Hammersley, 1974; Kingman, 1975; Semerjian et al., 2020)). *Consider the recursive distributional equation $K^{(t+1)} = \min_{1 \le i \le n} K_i^{(t)} + \Xi_i$, where $K_i^{(t)}$ and $\Xi_i$ are i.i.d copies of random variables $K_{(\cdot)}^{(t)}$ and $\Xi_{(\cdot)}$, we have $\frac{K^{(t+1)}}{t} \xrightarrow{\text{a.s.}} -\inf_{\theta > 0} \frac{1}{\theta} \log\left[\sum_{i=1}^n \mathbb{E} e^{-\theta \Xi_i}\right]$, conditional on the event $\lim_{t \to \infty} K^{(t)} \ne \infty$.*

With Theorem 1, we conclude the critical point for the correct permutation recovery, i.e., $H + H' > \Omega$, can be computed by letting $\inf_{\theta > 0} \frac{1}{\theta} \log\left[\sum_{i=1}^n \mathbb{E} e^{-\theta \Xi_i}\right] \le 0$, since otherwise the condition in (4) will be violated. In the oracle case where $\mathbf{B}^\natural$ is known, we have random variable $\Xi$ be written as

$$\Xi = \boldsymbol{x}^\top \mathbf{B}^\natural \mathbf{B}^{\natural\top} (\boldsymbol{x} - \boldsymbol{y}) + \sigma \boldsymbol{w} \mathbf{B}^{\natural\top} (\boldsymbol{x} - \boldsymbol{y}), \tag{9}$$

where $\boldsymbol{x}$ and $\boldsymbol{y}$ follow the distribution $\mathsf{N}(\mathbf{0}, \mathbf{I}_{p \times p})$, and $\boldsymbol{w}$ follows the distribution $\mathsf{N}(\mathbf{0}, \mathbf{I}_{m \times m})$.

For the convenience of computation, we consider the simple case where $\mathbf{B}^\natural$ is $\lambda \mathbf{I}_{p \times p}$ ($m = p$).

**Proposition 1.** *Consider the case where $\mathbf{B}^\natural$ is a re-scaled version of the identity matrix, i.e., $\lambda \mathbf{I}_{p \times p}$, we can write the expectation $\mathbb{E} e^{-\theta \Xi}$, which is defined in (9), as*

$$\mathbb{E} e^{-\theta \Xi} = \left(1 + 2\theta \lambda^2 - \theta^2 \lambda^2 \left(\lambda^2 + 2\sigma^2\right)\right)^{-\frac{m}{2}} \tag{10}$$

*provided that* $\qquad \theta^2 \sigma^2 \lambda^2 < 1, \quad \text{and} \quad \theta^2 \lambda^2 \left(\lambda^2 + 2\sigma^2\right) \le 1 + 2\theta \lambda^2.$ $\tag{11}$

Provided the conditions in (11) is violated, we have the expectation $\mathbb{E} e^{-\theta \Xi}$ to diverge to infinity, which suggests the optimal $\theta_*$ for $\inf_{\theta > 0} \log(n \mathbb{E} e^{-\theta \Xi})/\theta$ cannot be achieved. Using (10), we can compute the optimal $\theta_*$ as $\theta_* \cong \sqrt{\frac{2 \log n}{\|\mathbf{B}^{\natural\top} \mathbf{B}^\natural\|_\mathrm{F}^2 + 2\sigma^2 \|\mathbf{B}^\natural\|_\mathrm{F}^2}}$. The corresponding $\mathsf{snr}_{\text{oracle}}$ is written as

$$\mathsf{snr}_{\text{oracle}} \cong \frac{4 \log n}{m - 2 \log n}. \tag{12}$$

The comparison between the theoretical values of the phase transition threshold and the numerical values are put in Table 1, from which we conclude the phase transition threshold $\mathsf{snr}$ can be predicted to a good extent. As $m$ increases, we believe the gap between the theoretical values and the numerical values will keep shrinking.

Table 1: Comparison between the predicted value of the phase transition threshold $\mathsf{snr}_{\text{oracle}}$ and its numerical value when $n = 500$. **P** denotes the predicted value while **N** denotes the numerical value. **N** value corresponds to the $\mathsf{snr}_{\text{oracle}}$ when the error rate drops below 0.05.)

| $m$ | 20 | 30 | 40 | 50 | 60 | 70 |
|---|---|---|---|---|---|---|
| P | 3.283 | 1.415 | 0.902 | 0.662 | 0.523 | 0.432 |
| N | 2.466 | 1.290 | 0.862 | 0.644 | 0.513 | 0.426 |

## 4.2 APPROXIMATED COMPUTATION OF PHASE TRANSITION THRESHOLDS

The computation of phase transition threshold is by setting $\inf_{\theta > 0} \log\left(\mathbb{E} Z \cdot \mathbb{E} e^{-\theta \Xi}\right)/\theta$ be zero. However, in certain scenarios, it can be extremely difficult or impossible to obtain a closed-formula of $\mathbb{E} e^{-\theta \Xi}$, let alone the optimal solution $\theta$. To handle such difficulties, we propose to approximate $\mathbb{E} e^{-\theta \Xi}$ with Taylor expansion, which proceeds as

$$\mathbb{E} e^{-\theta \Xi} = e^{-\theta \mathbb{E}\Xi} \cdot \mathbb{E} e^{-\theta(\Xi - \mathbb{E}\Xi)} \overset{\text{①}}{=} e^{-\theta \mathbb{E}\Xi} \cdot \left[1 + \frac{\theta^2}{2} \cdot \mathbb{E}\left(\Xi - \mathbb{E}\Xi\right)^2 + \mathbb{O}_{\mathrm{P}}\left(\theta^4 \mathbb{E}\left(\Xi - \mathbb{E}\Xi\right)^4\right)\right],$$

where ① is due to the fact $\mathbb{E}\left(\Xi - \mathbb{E}\Xi\right)^3 = 0$. To simplify the computation, we adopt one widely-used assumption, stating as

**Assumption 1.** *We assume $\theta^4 \cdot \mathbb{E}\left(\Xi - \mathbb{E}\Xi\right)^4$ to be negligible.*

Then we obtain

$$\mathbb{E} e^{-\theta \Xi} \cong e^{-\theta \mathbb{E}\Xi} \cdot \left(1 + \frac{\theta^2}{2}\mathrm{Var}\Xi\right) \overset{\text{②}}{\cong} \exp\left(-\theta \mathbb{E}\Xi + \frac{\theta^2}{2}\mathrm{Var}\Xi\right), \tag{13}$$

where in ② we use the approximation $1 + x \cong e^x$ when $x$ is near zero. In this way, the rather complicated computation of $\mathbb{E} e^{-\theta \Xi}$ is replaced by the computation of mean $\mathbb{E}\Xi$ and variance $\mathrm{Var}\Xi$, which is still complex but manageable. With this approximation, the optimal $\theta_*$ for $\log\left(n E e^{-\theta \Xi}\right)/\theta$ is computed as $\sqrt{2 \log n / \mathrm{Var}\Xi}$ and hence the critical point corresponding to the phase transition as

$$2(\log n)\mathrm{Var}\Xi = \left(\mathbb{E}\Xi\right)^2. \tag{14}$$

To verify that this approximation can yield meaningful results, we revisit the oracle case and have

$$\mathbb{E}\Xi = \left\|\mathbf{B}^{\natural}\right\|_{\mathrm{F}}^2; \quad \mathrm{Var}\Xi = 3\left\|\mathbf{B}^{\natural}\mathbf{B}^{\natural\top}\right\|_{\mathrm{F}}^2 + 2\sigma^2\left\|\mathbf{B}^{\natural}\right\|_{\mathrm{F}}^2. \tag{15}$$

Plugging (15) into (14) then yields the relation

$$6 \log n\left\|\mathbf{B}^{\natural}\mathbf{B}^{\natural\top}\right\|_{\mathrm{F}}^2 + 4\sigma^2(\log n)\left\|\mathbf{B}^{\natural}\right\|_{\mathrm{F}}^2 = \left\|\mathbf{B}^{\natural}\right\|_{\mathrm{F}}^4,$$

from which we can determine the critical point of $\mathsf{snr}$.

**Discussion.** As a comparison, we first revisit the simple case where $\mathbf{B}^{\natural}$ is $\lambda \mathbf{I}_{p \times p}$ ($p = m$). We can compute the phase transition $\mathsf{snr}$ as $4 \log n / (m - 6 \log n)$. This solution is almost identical to (12) in the large-system limit as $\mathsf{snr}_{\text{oracle}} \cong \widetilde{\mathsf{snr}}_{\text{oracle}} \cong n^{\frac{4}{m}} - 1$. Moreover, we should stress that $(i)$ our approximation method applies to other types of matrices as well, rather than limited to the identity matrix; and $(ii)$ our approximation method can predict the phase transition thresholds even when the entries $\mathbf{X}_{ij}$ are sub-gaussian. An illustration is given in Table 2: in (**Case I**), half of eigenvalues are with Ener while the other half are with Ener/2; in (**Case II**), half of the eigenvalues are with Ener while the other half are with 3Ener/4.

## 5 ANALYSIS OF NON-ORACLE CASE

Having presented the oracle case as a warm-up example, we now extend the analysis to the non-oracle case, where the value of $\mathbf{B}^{\natural}$ is not given a prior. To begin with, we need to recast the permutation recovery problem as a LAP. As shown in Zhang et al. (2022), the ML estimator yields a *quadratic assignment problem* (QAP), which is NP-hard to solve and fails to meet this requirement. Fortunately,

Table 2: Comparison between the predicted value of the phase transition threshold $\mathsf{snr}_{\text{oracle}}$ and its numerical value when $n = 800$. **Gauss** refers to $\mathbf{X}_{ij} \overset{\text{i.i.d}}{\sim} \mathsf{N}(0,1)$ while **Unif** refers to $\mathbf{X}_{ij} \overset{\text{i.i.d}}{\sim}$ Unif$[-1,1]$. We averaged over 20 experiments.

| $m$ | 100 | 110 | 120 | 130 |
|---|---|---|---|---|
| (**Case I**) P | 0.63 | 0.52 | 0.43 | 0.38 |
| (Gauss) N | $0.6 \sim 0.65$ | $0.55 \sim 0.6$ | $0.55 \sim 0.6$ | $0.5 \sim 0.55$ |
| (Unif) N | $0.6 \sim 0.65$ | $0.6 \sim 0.65$ | $0.55 \sim 0.6$ | $0.55 \sim 0.6$ |
| (**Case II**) P | 0.47 | 0.40 | 0.35 | 0.31 |
| (Gauss) N | $0.5 \sim 0.55$ | $0.5 \sim 0.55$ | $0.45 \sim 0.5$ | $0.45 \sim 0.5$ |
| (Unif) N | $0.5 \sim 0.55$ | 0.5 | $0.45 \sim 0.5$ | $0.45 \sim 0.5$ |

the proposed estimator in Zhang & Li (2020) is able to fill the gap. Define the edge weight $\mathbf{E}_{ij}^{\text{non-oracle}}$ as $\mathbf{E}_{ij}^{\text{non-oracle}} \triangleq -\mathbf{Y}_i \mathbf{Y}^\top \mathbf{X} \mathbf{X}_j^\top$, we can reconstruct the permutation matrix $\mathbf{\Pi}^\natural$ as (1). Before proceeding, we would like to justify using the estimator in Zhang & Li (2020) to analyze the permuted linear regression in the non-oracle case: first, this estimator is proved to achieve both the computational and statistical optimality; second, this estimator also exhibits a phase transition phenomenon, which behaves similarly to that in the oracle case. Naturally, we should expect this estimator will incorporate some inherent properties of the permutation recovery problem (with unknown $\mathbf{B}^\natural$), from which we can gain meaningful insights.

## 5.1 ANALYSIS OF NON-ORACLE CASE

Having illustrated soundness of the approximation method in (13), we apply it to the non-oracle case, where the random variable is written as

$$\Xi = \Xi_1 + \sigma \left( \Xi_2 + \Xi_3 \right) + \sigma^2 \Xi_4, \tag{16}$$

where $\Xi_i$ $(1 \le i \le 4)$ are defined as

$$\Xi_1 \triangleq \mathbf{X}_{\pi^\natural(i)}^\top \mathbf{B}^\natural \mathbf{B}^{\natural\top} \mathbf{X}^\top \mathbf{\Pi}^{\natural\top} \mathbf{X} \left( \mathbf{X}_{\pi^\natural(i)} - \mathbf{X}_j \right); \qquad \Xi_2 \triangleq \mathbf{X}_{\pi^\natural(i)}^\top \mathbf{B}^\natural \mathbf{W}^\top \mathbf{X} \left( \mathbf{X}_{\pi^\natural(i)} - \mathbf{X}_j \right);$$

$$\Xi_3 \triangleq \mathbf{W}_i^\top \mathbf{B}^{\natural\top} \mathbf{X}^\top \mathbf{\Pi}^{\natural\top} \mathbf{X} \left( \mathbf{X}_{\pi^\natural(i)} - \mathbf{X}_j \right); \qquad \Xi_4 \triangleq \mathbf{W}_i^\top \mathbf{W}^\top \mathbf{X} \left( \mathbf{X}_{\pi^\natural(i)} - \mathbf{X}_j \right),$$

respectively. Then we conclude

**Theorem 2.** *The mean $\mathbb{E}\Xi$ of $\Xi$ in (39) and its variance $\mathrm{Var}\Xi$ are computed as*

$$\mathbb{E}\Xi \approxeq n \left( 1 - \tau_h \right) \left[ \left( 1 + \tau_p \right) \left\| \mathbf{B}^\natural \right\|_{\mathrm{F}}^2 + n \tau_m \tau_p \sigma^2 \right];$$

$$\mathrm{Var}\Xi \approxeq n^2 \tau_h \left( 1 - \tau_h \right) \tau_p^2 \left[ \left\| \mathbf{B}^\natural \right\|_{\mathrm{F}}^2 + m \sigma^2 \right]^2 + n^2 \left[ 2\tau_p + 3 \left( 1 - \tau_h \right)^2 \right] \left\| \mathbf{B}^{\natural\top} \mathbf{B}^\natural \right\|_{\mathrm{F}}^2$$

$$+ n^2 \left[ 6\tau_p \left( 1 - \tau_h \right)^2 + \left( 3 - \tau_h \right) \tau_p^2 \right] \left\| \mathbf{B}^{\natural\top} \mathbf{B}^\natural \right\|_{\mathrm{F}}^2,$$

*respectively, where the definitions of $\tau_p$, $\tau_m$ and $\tau_h$ can be found in Section 2.*

For the clarity of presentation, we only present the proof outlines.

**Computation of mean $\mathbb{E}\Xi$.** For the computation of the mean $\mathbb{E}\Xi$, easily we can verify that $\mathbb{E}\Xi_2$ and $\mathbb{E}\Xi_3$ are both zero, which is due to the independence between $\mathbf{X}$ and $\mathbf{W}$. Regarding the computation of $\mathbb{E}\Xi_1$ and $\mathbb{E}\Xi_4$, we adopt Wick's theorem and obtain

$$\mathbb{E}\Xi_1 = n \left( 1 - \tau_h \right) \left( 1 + \tau_p \right) \left[ 1 + o_{\mathsf{P}} \left( 1 \right) \right] \left\| \mathbf{B}^\natural \right\|_{\mathrm{F}}^2; \quad \mathbb{E}\Xi_4 = n^2 \tau_m \tau_p \left( 1 - \tau_h \right) \left( 1 + o_{\mathsf{P}} \left( 1 \right) \right).$$

**Computation of variance $\mathrm{Var}\Xi$.** With the relation $\mathrm{Var}(\Xi) = \mathbb{E}\Xi^2 - (\mathbb{E}\Xi)^2$, our goal becomes computing $\mathbb{E}\Xi^2$, which consists the calculation of the following six terms

$$\mathbb{E}\Xi^2 = \mathbb{E}\Xi_1^2 + \sigma^2 \mathbb{E}\Xi_2^2 + \sigma^2 \mathbb{E}\Xi_3^2 + \sigma^4 \mathbb{E}\Xi_4^2 + 2\sigma^2 \mathbb{E}\Xi_1\Xi_4 + 2\sigma^2 \mathbb{E}\Xi_2\Xi_3.$$

The computation of above terms turns to be quite complex due to the high order Gaussian chaos. For example, term $\mathbb{E}\Xi_1^2$ involves the eighth-order Gaussian chaos; terms $\mathbb{E}\Xi_2^2, \mathbb{E}\Xi_3^2, \mathbb{E}\Xi_1\Xi_4$ and $\mathbb{E}\Xi_2\Xi_3$ all involves the sixth-order Gaussian variables. To alleviate the computational burden, we compute the expectation $\mathbb{E}\Xi^2$ in the following three phases.

Table 3: Comparison between the predicted value of the phase transition threshold $\tau_h$ and its numerical value when $n = 500$. **P** denotes the predicted value while **N** denotes the numerical value. (The numerical value of $\tau_h$ is the minimum $\tau_h$ when the correct permutation rate drops blow 0.05.)

| $p$ | 75 | 100 | 125 | 150 | 175 | 200 |
|---|---|---|---|---|---|---|
| P | 0.82 | 0.73 | 0.68 | 0.62 | 0.56 | 0.52 |
| N | 0.77 | 0.74 | 0.7 | 0.66 | 0.61 | 0.57 |

- **Phase I.** The solution in this phase comes from a modification of the so-called *leave-one-out* technique (Sur et al., 2019; El Karoui, 2013; 2018; Bai & Silverstein, 2010). Notice that the major technical difficulty comes from the correlation between the product $\mathbf{X}^\top \mathbf{\Pi}^\sharp \mathbf{X}$ and the difference $\mathbf{X}_{\pi^\sharp(i)} - \mathbf{X}_j$. We decompose this correlation by first rewriting the matrix $\mathbf{X}^\top \mathbf{\Pi}^\sharp \mathbf{X}$ as the sum $\sum_\ell \mathbf{X}_\ell \mathbf{X}_{\pi^\sharp(\ell)}^\top$. Then we collect all terms $\mathbf{X}_\ell \mathbf{X}_{\pi^\sharp(\ell)}^\top$ independent of $\mathbf{X}_{\pi^\sharp(i)}$ and $\mathbf{X}_j$ in the matrix $\mathbf{\Sigma}$ and leave the rest terms to matrix $\mathbf{\Delta}$, which means $\mathbf{\Delta} \triangleq \mathbf{X}^\top \mathbf{\Pi}^\sharp \mathbf{X} - \mathbf{\Sigma}$. This decomposition is in the same spirit of the leave-one-out technique.

  With this method, we divide all terms involved in the computation of $\mathbb{E}\Xi^2$ into three categories: $(i)$ those only containing matrix $\mathbf{\Sigma}$; $(ii)$ those containing both $\mathbf{\Sigma}$ and $\mathbf{\Delta}$; and $(iii)$ those only containing $\mathbf{\Delta}$. Easily we can see that the first two categories contain most vectors' outer products while the last category only contains a finite number of such terms.

- **Phase II.** Concerning the terms in the first two categories, which contains majority of terms, we can exploit the independence among rows in the sensing matrix $\mathbf{X}$ and reduce the order of Gaussian random variables by separately taking expectation w.r.t $\mathbf{\Sigma}$ and w.r.t vectors $\mathbf{X}_{\pi^\sharp(i)}$ and $\mathbf{X}_j$.

- **Phase III.** For the few terms in the third category which contains high-order Gaussian chaos, we compute their expectations by iterative applying of Wick's Theorem and Stein's Lemma, which filters out the zero terms to reduce higher-order interactions between Gaussian random variables to lower-order interactions.

For more technical details, we refer the interested readers to the supplementary material.

## 5.2 IDENTIFYING THE PHASE TRANSITION THRESHOLD

Having explained the computation of $\mathbb{E}\Xi$ and $\mathrm{Var}\Xi$, we turn to identifying the phase transition threshold. Different from the oracle case, we notice the edge weight $E_{ij}$ are strongly correlated in this case especially when $j = \pi^\sharp(j)$, which corresponds to the non-permuted rows. To factor out these independence, we only take the permuted rows into account and correct the sample size from $n$ to $\tau_h n$. Thus, we have

**Proposition 2.** *The critical point for the phase transition phenomenon transforms from* (14) *to* $2 \left( \log \tau_h n \right) \mathrm{Var}\Xi = \left( \mathbb{E}\Xi \right)^2$.

**Example 1.** *We consider the case where* $\mathbf{B}^\sharp = \lambda \mathbf{I}_{p \times p}$ *as an illustration. With Theorem* 2 *and Proposition* 2, *we obtain the solution*

$$\mathsf{snr}_{\text{non-oracle}} = \eta_1/\eta_2, \tag{17}$$

*where* $\eta_1$ *and* $\eta_2$ *are defined as*

$$\eta_1 \triangleq 2\tau_h \tau_p^2 \log\left(n\tau_h\right) - \tau_p(\tau_p + 1)\left(1 - \tau_h\right) + \sqrt{2}\tau_p \sqrt{(1 - \tau_h)\tau_h \left( \log\left(n\tau_h\right) \right)};$$
$$\eta_2 \triangleq 2\tau_h \tau_p^2 \log(n\tau_h) - \left(1 - \tau_h\right)\left(\tau_p + 1\right)^2.$$

*Notice that the negative solution has been abandoned due to the non-negativity requirement of* $\mathsf{snr}$.

***Discussion.*** *For the accuracy of the predicted phase transition* $\mathsf{snr}_{\text{non-oracle}}$, *we notice a increasing gap between the theoretical value and the numerical value when compared with that in the oracle case. Possible reasons include strong correlation across the edge weights* $\{E_{ij}\}_{1 \leq i,j \leq n}$ *and the error within the approximation relation* $\mathbb{E}e^{-\theta\Xi} \approx \mathbb{E}\exp\left(\theta\mathbb{E}\Xi - \theta^2 \mathrm{Var}\Xi/2\right)$.

*In addition, we observe a singularity point, i.e.,* $\tau_h$ *is approximately* 0.73 *in Figure* 2, *which suggests a phase transition phenomenon. To validate the predicted phenomenon, we consider the noiseless*

*case, i.e.,* $\mathsf{snr} = \infty$, *and reconstruct the permutation matrix* $\mathbf{\Pi}^{\natural}$ *with* (1). *Numerical experiments confirm our prediction by showing that the correct rate of the permutation recovery exceeds* $0.2$ *when* $h/n \leq 0.73$ *and drops below* $0.05$ *when* $h/n > 0.74$. *Additional experiments are put in Table* 3, *from which we conclude the solution* (17) *can predict the critical points w.r.t.* $\tau_h$ *to a good extent.*

**Remark 1.** *Compared with the prior work (Zhang & Li, 2020) which only yields the statistical order, i.e.* $h/n \leq c$ *(c is a positive constant), our framework can (i) specify the positive constant c, and (ii) uncover the dependence of* $\tau_h$ *on the ratio* $\tau_p$. *In addition, Zhang & Li (2020) requires* $n \gg p$ *while our work allows n to be the same order of p, i.e.,* $p/n = \tau_p$ *as* $n, p \to \infty$. *Notice that this is consistent with numerical experiments such that* $n = \mathbb{O}_{\mathsf{P}}(p)$ *is sufficient for the permutation recovery.*

## 6 PARTIAL PERMUTATION RECOVERY

Apart from the phase transition thresholds, we would like to exploit MP to design algorithms for a partial permutation recovery. As an illustration, we consider recovering the correspondence $\pi^{\natural}(i)$ for a single index $i$. In an effortless way, we can generalize it to recovering the correspondences for multiple indices by iterative applying the following procedure.

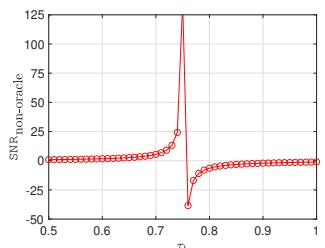

Figure 2: $\mathsf{snr}_{\text{non-oralce}}$ when $n = 500$ and $p = 100$.

To start with, we modify the graphical model in Figure 1 by removing the unnecessary function nodes and their incident edge. Then we can rewrite the MP update equation

$$\widehat{m}_{i^{\mathsf{L}} \to (i^{\mathsf{L}}, j^{\mathsf{R}})}(\pi) \simeq \sum_{\pi_{i^{\mathsf{L}}, k^{\mathsf{R}}}} \prod_{k^{\mathsf{R}} \neq j^{\mathsf{R}}} \exp\left(-\beta \pi_{i^{\mathsf{L}}, k^{\mathsf{R}}} E_{i^{\mathsf{L}}, k^{\mathsf{R}}}\right) \mathbb{1}(\pi + \sum_k \pi_{i^{\mathsf{L}}, k^{\mathsf{R}}} = 1).$$

Compared with the MP for the full permutation recovery, MP for the partial permutation recovery can reach convergence in one iteration. Letting $\beta \to \infty$, we obtain the edge selection criteria as

$$\widehat{\pi}(i^{\mathsf{R}}) = \operatorname{argmin}_j E_{ij},$$

which turns out to be a greedy selection scheme.

Following the same procedure as in Sections 4 and 5, we can analyze its statistical properties, which are similar (although degraded) to the estimator for the full permutation recovery thereof (both the oracle case and non-oracle case). This claim is also confirmed by numerical experiments in Figure 3.

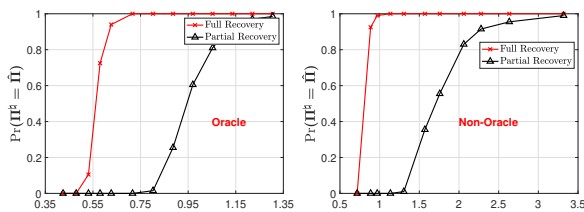

Figure 3: Comparison between the full permutation recovery and partial permutation recovery. We set $n = 500$, $p = 100$, and $m = 75$. The partial permutation recovery estimator also exhibits a phase transition, similar to the full permutation recovery estimator. However, their phase transition points are larger when compared with the corresponding points for the full permutation recovery.

## 7 CONCLUSION

This is the first work that can identify the precise location of phase transition thresholds of permuted linear regressions. For the oracle case where the signal $\mathbf{B}^{\natural}$ is given a prior, our analysis can predict the phase transition threshold $\mathsf{snr}_{\text{oracle}}$ to a good extent. For the non-oracle case where $\mathbf{B}^{\natural}$ is not given, we modified the leave-one-out technique to approximately compute the phase critical $\mathsf{snr}_{\text{non-oracle}}$ value for the phase transition, as the precise computation becomes significantly complicated as the high-order interaction between Gaussian random variables is involved. Moreover, we associated the singularity point in $\mathsf{snr}_{\text{non-oracle}}$ with a phase transition point w.r.t the maximum allowed number of permuted rows. In the end, we generalized the full permutation recovery and obtained a partial permutation recovery algorithm. Following the same analytical procedure, we argued it would have a similar although degraded performance compared with the full permutation recovery algorithm, which is later confirmed by our numerical experiments. In the future, we will incorporate the *replica symmetry breaking* scheme into our framework and extend this framework to broad areas.

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
