# OpenReview forum: "Identifying Phase Transition Thresholds of Permuted Linear Regression via Message Passing"
_ICLR.cc/2023/Conference — Submitted to ICLR 2023_

### Official Review · Reviewer_eTFJ · 2022-10-17

**Confidence:** 4
**Correctness:** 1
**Technical Novelty And Significance:** 3
**Empirical Novelty And Significance:** Not applicable
**Recommendation:** 1

**Clarity, Quality, Novelty And Reproducibility:**

The major weakness of the paper is that its mathematical writing is so
terrible that a general ICLR reader (me for example) is not able to
understand the paper. From this, I conclude the score of reject.
Many examples of inaccuracies in writing are given below.

1. Section 3 is titled "Background Knowledge on Graphical Models", but
   it starts with a review of the linear assignment problem. In (1),
   the global minimizers might not be unique, so it should be $\in$.
   This is very important, because the authors should make it clear
   whether their theorems hold for any global minimizers of (1), or
   for some particular global minimizer, or there exists a global
   minimizer of (1) for which the theorems hold.
2. I got confused as early as at equation (2).
   What is the indicator function there? The standard definition should
   involve a set, as in the wikipedia page,
   <https://en.wikipedia.org/wiki/Indicator_function>). Why does this
   define a probability measure over permutations? Why the ML
   estimator maximizes this measure? Where is the ML estimator
   defined? What are the marginals of this measure?
3. The review of the message passing (MP) algorithm in Sections 3.1 and 3.2 is also
   confusing. This could be criticized from several points:
   - The last two paragraphs of Section 3.1 reviewed the MP
     algorithm, then a reader found the title of Section 3.2
     is the MP algorithm. This indicates that the last two paragraphs
     are misplaced.
   - The last two paragraphs are very confusing. What does it mean to
     say **multiplying all incoming messages**? The authors refer the
     readers to a book of more than 500 pages for an introduction to
     MP (Mezard and Montanari, 2019). Are the authors expecting the
     readers to take a year for studying the MP algorithm and understand the
     paper? In the last paragraph, the authors suggest that applying
     the MP algorithm to their case might yield invalid results, and
     then that doing so might yield meaningful results. The readers have
     no idea what that means.
   - Section 3.2 is a disaster for any readers. Let us start the
     criticism from the first paragraph. There, the authors simply
     denote the **message flow** by a complicated symbol, without
     explaining what the **message flow** is. The notation about
     **left** and **right** is not informative as to what it means, eg
     what is a left variable node? The message flow is sometimes
     defined by $m$, and sometimes by $\hat{m}$, for which the
     motivation is not very clearly explained. Finally, the authors
     repeatedly refer the readers to a 500+ pages book.
   - When reading the rest part of Section 3.2, I am completely
     lost. What does the equality symbol mean (which is a combination of $\approx$
     and $=$) in (3)? What does it mean to say $\pi\in\{0,1\}$ is a binary data?
     It was previously defined as some permutation map. The sentence
     ``Similarly, we can write ...'' has a serious grammar problem, an
     so is the sentence between (4) and (5).
4. Section 4 is written in a sub-optimal way. The reasons are listed below.
   - Why (6) is the ML estimator?
   - $E^{oracle}$ could be defined more straightforwardly by
     saying that the definition of $Y$ is used. However, The
     definition of $E^{oracle}$ is rarely useful for the reader and
     for later development, as (6) is already an instance of the
     LAP problem. This definition is useful for the purpose of proof.
   - Now we look at Section 4.1. In equation (9), a
     weird quantity is defined, and the reader does not even know how
     to pronounce this symbol. The third paragraph of this section
     starts with **This equation**, but the reader never knows which
     equation it is. Grammar mistakes are found in the sentences below
     (11) and (12).
   - Two inaccuracies are found in Section 4.2. Assumption 1 is not
     rigorous, what does it mathematically mean to say a quantity is
     negligible? In the last paragraph, what is **Ener**?
5. The authors and reviewers would guess that I am now going to talk
   about Section 5. However, I give it up. I wish I have collected
   enough evidence for my claim that the paper is poorly written. In
   fact, Section 5 heavily relies on previous sections, and it is thus
   not hard to imagine its quality in writing.
6. Generally speaking, one of the major problems is that the authors throw a bunch of
   definitions, notations, and conditions to the readers, without even
   trying to explain them. One outstanding example is in Equation
   (16), where the weird symbol is decomposed into several ones, each
   one associated with a complicated definition. Yet, explanations for
   them are entirely missing, which are excellent only for testing how clever
   the readers are.

**Strength And Weaknesses:**

# Weaknesses

1. Proposition 1 relies on an unrealistic assumption where $B$ is a rescaled identity matrix.
2. The paper contradicts itself by arguing two converse points. On the one hand, the authors congratulate on their identifying the **precise locations** of the phase transition phenomenon for the problem of interest. On the other, the title of Section 4.2 indicates that the phase transition threshold is computed **approximately**.
3. The authors argue that they propose **the first framework** for something. The word **framework** is so vague that it does not make sense to me. If it means the message-passing framework, it has already been known.
4. The authors argue that the proposed partial permutation estimator is a **generalization** of the full permutation estimator. This claim is not supported by any explanations.
5. The authors argue that **the empirical phase transition points are well aligned with theoretical predictions**. However, the theoretical results are asymptotic, and the empirical results are for finite samples. There is a huge gap between practice and theory, and this claim amounts to aligning apples with oranges.

**Summary Of The Paper:**

In this paper, the authors studied the permuted linear regression problem using the message-passing framework.

The authors argue that they identify the **precise locations** of the phase transition phenomenon in this problem.

The analysis is divided into two cases, the case where the ground-truth B is given (the oracle case), and the case where it is not.

Finally, the authors propose an estimator for partial permutation recovery.

**Summary Of The Review:**

I am not an expert in the field. Thus I did not evaluate the technical contributions.

However, my judgment is that the paper makes a lot of claims that are not well supported by theory or experiments, and the writing is highly sub-optimal. This renders the paper inaccessible to an ordinary ICLR reader and inappropriate for ICLR publications.

I recommend the authors make a significant revision and consider resubmission to other avenues (e.g., for statisticians or physicians). Actually, the paper might have a better fate if submitted to AISTATS.

---

> ### Author Response · Authors · 2022-11-19
> **Thank you for your review**
>
> Dear Reviewer,
>
> we appreciate your criticism on our work, which will certainly motivate us to find better ways to convince a broader range of audience.  Also thanks for admitting the very low score is not based on evaluating of our technical contribution: "*I am not an expert in the field. Thus I did not evaluate the technical contributions.*"
>
> We hope to reply to the "weakness" listed in your review
>
>
> **1. Proposition 1 relies on an unrealistic assumption where $B$  is a rescaled identity matrix.**
>
> We consider a special case for sake of easy comparison, which is stated one line above Proposition 1. Our method can be applied to other types of matrices effortlessly.
>
>
> **2. The paper contradicts itself by arguing two converse points. On the one hand, the authors congratulate on their identifying the precise locations of the phase transition phenomenon for the problem of interest. On the other, the title of Section 4.2 indicates that the phase transition threshold is computed approximately.**
>
> The *computed approximately* means we adopt some  approximations in the derivation; while the *precise location* means our obtained results are close to the experiment results. In other words, this suggests our approximation does not bring  significant errors.
>
>
>
> **3 The authors argue that they propose the first framework for something. The word framework is so vague that it does not make sense to me. If it means the message-passing framework, it has already been known.**
>
> We mean that we first introduce the message-passing algorithm to the permuted linear regression research.
>
>
> **4 The authors argue that the proposed partial permutation estimator is a generalization of the full permutation estimator. This claim is not supported by any explanations.**
>
> The partial recovery is derived by 1) removing the unwanted nodes in the graphical modes and their connecting edges; and 2) applying the message-passing algorithm. This is explained in Page 9 (left side of Figure 2).
>
>
> **5 The authors argue that the empirical phase transition points are well aligned with theoretical predictions. However, the theoretical results are asymptotic, and the empirical results are for finite samples. There is a huge gap between practice and theory, and this claim amounts to aligning apples with oranges.**
>
> This is quite a unusual comment. Typically, people use asmptotics to derive closed-form expression and validate that their asymptoic expressions also match the finite-sample performance. When asymptoic expression matches finite sample behavior, people tend to celebrate the success. We agree it is also important to derive closed-form exact expressions for finite samples, but often not possible.

---

### Official Review · Reviewer_PjJn · 2022-10-24

**Confidence:** 3
**Correctness:** 3
**Technical Novelty And Significance:** 3
**Empirical Novelty And Significance:** Not applicable
**Recommendation:** 6

**Clarity, Quality, Novelty And Reproducibility:**

The quality of the results is good, the authors cleverly exploit techniques developed for a planted graph matching problem to tackle a permuted linear regression one.

The precise location of the phase transitions is a novel contribution since prior works were only concerned with the statistical order. Moreover the analysis allows to study the intertwined influence among the different aspect ratios in the problem which was not studied in detail before this work.

The authors do not mention any future release of the code which affects the reproducibility of the results.

The work is correctly structured, however I think that the clarity in the presentation should be improved. I have the following comments:
#### Questions:
- *Section 1.:*
"[...]  we also come up with an algorithm  to partially recover the permutation matrix [...]" how does this result frame in the literature of partial recovery?
- *Section 4.1:*
 a) Is there a way to build intuition on the equations defining the phase transition (10,11)?; b) What are the computational limitations in your work? Why in Table 1 you could not analyze higher $m$?
- *Section 4.2:*
a) I do not think that Table 2 alone is enough to conjecture that the method works also for Sub-Gaussian matrices. Is it just a numerical observation or is there an analytical argument? b) Define Ener in Table 2.
 - *Section 5:"[...]first this estimator is proven to achieve statistical optimality, second exhibit phase transition[...]"* in which work these claims are proven?
- *Section 5.2:*
 a) Why do you exclude from the possible reasons for the approximation failure the fact that you neglected the edges $j = \pi(j)$? Considering $\tau_h n$ instead of $n$ as sample size is exact?
 b) The caption of Figure 2 is not fully clear: are the points simulations?
- *Section 6:*
a) What are related works on partial permutation recovery? How do the results of this section frame into broader context? b) Is there a theoretical prediction in figure 3 or is just MP?; c) Is the analysis of the statistical properties of the estimator for partial recovery reported somewhere?
- *Section 7:*
 a) What are the main limitations of the setting studied? ; b) What would be the advantage of including RSB effect? It would be interesting to discuss more in detail these points.
#### Minor issues:
- *Abstract - "As is shown in the previous work..."* should be "As is shown in previous works"?. In the same sentence I would also avoid to use "etc." while explaining what are the relevant parameters that tune the phase transitions.
- *Abstract - "[...]via the message passing[...] "* should be "via a message passing"?
- *Section 1. - "[...]recovery suddenly drops to zero once some parameters[...]"* could become "once the relevant parameters"
- *Section 3.: "[...]ML estimator[...]"* use full name Maximum-Likelyhood since is the first time it is mentioned.
- I think that "A prior" should be changed with "a priori" in some points of the presentation.
- *Section 7:* "although degraded" should be in parenthesis



**Strength And Weaknesses:**

The paper is generally pleasant to read and the introduction offers an interesting broad picture of the field. The precise characterization of the snr threshold for the full recovery of $\Pi$ is a novel contribution. The details of the technical discussion are properly explained, indeed both the construction of the MP algorithm and the approximation method are discussed thoroughly. The analysis of the distributional equations via the connection with a BRW process is correctly reported.


The comparison of theoretical and algorithmic prediction is done in different scenarios and accompanied with numerical simulations, however the clarity in the presentation of the results could be improved.
The proposed approximation framework to pinpoint the threshold is interesting, although the mapping to a BRW process has been already presented in *Semerjan et al. 2020*, which marginally limts the originality of the work. There is almost no discussion about the limitations of this work, e.g. the restricting assumption about additive Gaussian noise, and the motivations for future directions of research are briefly reported.

**Summary Of The Paper:**

This work explores the high dimensional limit of a permuted linear regression problem with Gaussian sensing matrix and additive Gaussian noise.  Thanks to the introduction of a tailored graphical model, the authors build a Message Passing (MP) algorithm to reconstruct the permutation matrix $\Pi$. The theoretical analysis is focused on the precise characterization of the phase transition for perfect recovery of $\Pi$. The authors first study the oracle case, in which the signal of interest is known, and secondly the non-oracle scenario. The snr threshold is predicted analytically by writing recursive distributional equations and exploiting a mapping to a branching random walk process; an approximation framework is proposed to deal with untractable cases. Numerical experiments which highlight the phase transitions are presented, and the theoretical predictions are close to the algorithmic ones.

**Summary Of The Review:**

My score is due to the nice findings presented in this submission. My main concerns are about the lack of clarity in the presentation and the reproducibility of the results in this submission. The present grade is dependent on the future release of the code.

---

> ### Author Response · Authors · 2022-11-19
> **Thank you for your review**
>
> Dear Reviewer,
>
> We highly appreciate your encouraging and constructive comments, which we will incorporate in the revision. Thanks for letting us know that you hope to see the code. We will be happy to release the code to the public.

---

### Official Review · Reviewer_K1xv · 2022-10-25

**Confidence:** 2
**Correctness:** 4
**Technical Novelty And Significance:** 3
**Empirical Novelty And Significance:** Not applicable
**Recommendation:** 6

**Clarity, Quality, Novelty And Reproducibility:**

Clarity and Quality: The paper is relatively clearly written with some typos.

Novelty: I think the paper introduces some technical novelty, but at a higher level, the techniques are heavily inspired by prior work.

Reproducibility: As far as I can tell, the authors do not make the code for their paper public.



**Strength And Weaknesses:**

Strengths:
1. I think the paper proves an interesting result, demonstrating the phase transitions that occur for this problem.

Weaknesses:
1. The paper has a several typos, it might help to put the prose through a grammar and spell checker.
2. The techniques used in the paper seem standard.


**Summary Of The Paper:**

This paper considers the problem of permuted linear regression. Here, you are given $Y \in \mathbb R^{n \times m}, X \in \mathbb R^{n \times p}$ such that $Y = \Pi X B + W$ where $\Pi$ is an unknown permutation matrix, $B \in \mathbb R^{p \times m}$ is the signal matrix, $X \in \mathbb R^{n \times p}$ is the set of Gaussian measurements and $W$ is additive Gaussian noise.  From these, the goal is to reconstruct the unknown permutation matrix.

This paper focuses on identifying the phase transitions that occur for this problem, in terms of recoverability, as the parameters tend to infinity with ratios of the relevant parameters tending to some bound. They attempt to predict the phase transition in the limit by investigating the behaviour of a message passing algorithm.



**Summary Of The Review:**

Overall, I think this is an interesting paper that attempts to predict the relationships between various parameters in the large-system regime, at the points of phase transition (specifically, they predict the snr at which the phase transition occurs). Unfortunately I am not extremely familiar with the literature surrounding message passing algorithms, and so my confidence is low.

---

> ### Author Response · Authors · 2022-11-19
> **Thank you for your review**
>
> Dear Reviewer,
>
> Thank you very much for your encouraging comments and for your appreciation of our research results. Indeed, permutation recovery is a difficult and interesting research problems with potential applications in databases, privacy, and security. Message-passing is also very interesting, although it has been criticised for being non-rigorous. From our perspective, if we can derive interesting expressions to explain  useful experimental phenomenon, it is enough motivation to pursue the research.
>
> Thank you for pointing out typos and suggesting us to release the code to public. We will be happy to comply.

---

### Official Review · Reviewer_vjWT · 2022-10-27

**Confidence:** 3
**Correctness:** 2
**Technical Novelty And Significance:** 2
**Empirical Novelty And Significance:** 2
**Recommendation:** 3

**Clarity, Quality, Novelty And Reproducibility:**

This paper is not well-written and some parts are difficult to follow.

The technical results for the oracle case seem to be a corollary of Theorem 1, which is well-established in prior works. The theoretical results for the non-oracle case are not fully rigorous, and the authors are only able to approximately identify the phase transition threshold $\mathrm{snr}_{\mathrm{non-oracle}}$.

For reproducibility, the authors are suggested to include the code for their numerical results into the supplementary material.

**Strength And Weaknesses:**

Strength:

A precise identification of the phase transition thresholds for permuted linear regression would be of interest.

Weaknesses:

As far as I can tell, the major contribution is the derived phase transition threshold $\mathrm{snr}_{\mathrm{non-oracle}}$ for the non-oracle case.  In particular, the assumption about a known signal matrix seems to be not practically meaningful (and it is mentioned at the beginning of Section 4 that it is a "warm-up" example), and the analysis seems to be simply a corollary of the well-established Theorem 1 in prior works. In addition, from this submission, I cannot see the practical motivation for considering partial permutation recovery instead of full permutation recovery. (The authors claimed on page 2 for the partial permutation estimator that "we show its performance almost match the estimator for the full permutation recovery". But this claim is clearly overstated according to Figure 3 which shows that partial recovery clearly requires significantly larger snr).


Moreover, the writing of the paper is somewhat messy or non-rigorous:

- The authors mentioned on page 2 that "In the non-oracle case where $\mathbf{B}^♮$ is not given, our scheme can further predict the maximum allowed permuted rows and uncover its dependence on the ratio $n/p$". Such a sentence is a bit strange. If the authors want to include the results for the simpler oracle case in the main document, it is better to also include the prediction for the maximum allowed permuted rows and the uncovering of its dependence on the ratio $n/p$ for the oracle case.

- Assumption 1 is weird. The authors should provide justifications for it (and provide a rigorous characterization of "to be negligible"), instead of saying that it is a widely-used assumption.

- It is difficult for me to follow how the authors go from Theorem 2 to Proposition 2. In addition, in Example 1, the case that $\mathbf{B}^♮$ seems to be not practical. I hope that there are more discussions on Proposition 2 and the corresponding examples.

- In Table 2, for the numerical values, $0.6 \sim 0.65$ is a strange form of representation. It should be $\mathrm{mean} \pm \mathrm{std}$. Similar modifications should also apply to the numerical values in Table 1.

- $\mathbb{O}_P(\cdot)$ is used on page 1 before its definition in the Notations section. The authors should mention that $\mathbf{W}_i$ represents the $i$th row of $\mathbf{W}$. In Eq. (6), the constraints can be simplified as $\mathbf{\Pi} \in \mathcal{P}_n$, which has been defined in the Notations section.

- In the Discussion on page 6, $\tilde{snr}_{\mathrm{oracle}}$ appears suddenly (it seems to be a redundant notation) and what does $\mathrm{Ener}$ refer to? In the statement of Theorem 2, the referred Eq. (39) is in the supplementary material (and in fact, it should be Eq. (40)). It is better to write down Eq. (40) explicitly in the main document.

**Summary Of The Paper:**

This paper studies the phase transition thresholds of permuted linear regression in terms of the signal-to-noise ratio (snr). The authors first consider the oracle case where the signal of interest is known and predict the phase transition threshold $\mathrm{snr}_{\mathrm{oracle}}$.

Then, they study the non-oracle case where the signal of interest is not given and approximately identify the phase transition threshold $\mathrm{snr}_{\mathrm{non-oracle}}$ based on a modification of the leave-one-out technique. In addition, the full permutation estimator is generalized to a partial permutation estimator in Section 6.

**Summary Of The Review:**

Based on all the above comments, my impression is that this submission is not ready for publication in the current version.

---

> ### Author Response · Authors · 2022-11-19
> **Thank you for your review**
>
> Dear Reviewer,
>
> We thank you for the feedback and summary of strength *"A precise identification of the phase transition thresholds for permuted linear regression would be of interest."*
>
> We admit this is not an easy paper to write and to read because it involves two (to many researchers) difficult topics: 1) message-passing; 2) permutation recovery. On the other hand, we hope this also means the presented research might be interesting.
>
> Thank you for mentioning that message-passing lacks rigor (for general graphs). This is a known issue of the message-passing literature, which is probably the reason why many papers using message-passing methods have been published in physics venues. For our work, we derive the expression which matches the experiments for the phase-transition phenomenon that is interesting to many.
>
> In the meanwhile, we will of course try our best to incorporate your comments in the revision, although we do not expect that we will be able to make message-passing a fully rigorous method, at least not in this paper. Hopefully ICLR is a venue that welcomes works which derive interesting expressions to explain experiments.
>
>
>
> As for the performance comparison between full recovery and partial recovery, we have to mention that the custom of permutation recovery
> is to evaluate the performance in terms of $\log snr$.
> Examples include pananjady et al. 2017, zhang et al. 2020, slawski & ben-david 2019, etc.
> Adopting this metric, the curves of the full recovery
> and partial recovery almost coincide with each other.
> Here we have to specifically use SNR in the X-axis to highlight their differences.
>
>
>
> Thanks again for the valuable input.

---

### Official Review · Reviewer_tsTr · 2022-11-02

**Confidence:** 3
**Correctness:** 3
**Technical Novelty And Significance:** 3
**Empirical Novelty And Significance:** 3
**Recommendation:** 3

**Clarity, Quality, Novelty And Reproducibility:**

The paper is notationnally heavy and requires the careful reading of other related papers for a proper understanding (For someone who has not read Semerjian et al, the exposition and the notations are quite difficult to follow). The paper should be self contained which it clearly isn't (at least in its current state). I do believe there is more space required for a clear exposition of the mathematics. My suggestions, if you really want to submit to a conference, would be to remove part of the introduction (which can be found exactly in Semerjian et al.) as well as the proof sketch for Theorem 2 (which is unclear). Then use the spared space to expand on the connection between the oracle and the non oracle settings, discuss the experiments in more details and develop on the non-oracle estimator from Zhang et al 2020 which is a key element of the machinery.





**Strength And Weaknesses:**

The paper is interesting although poorly written. I have some concern with (1) the incremental nature of the result; Although there is a clear mathematical venture/achievement and the necessary associated hard work, the novelty lies in part in an approximation (based on a Taylor expansion) used to compute the phase transition when considering the non oracle setting; and perhaps more importantly (2) the organization/intelligibility of the paper.  The first part of the paper is very similar to Semerjian et al 2020 and the second part is an analysis based on the estimator provided in Zhang et al 2020. A reader unfamiliar with the former paper will have a very difficult time going through the notations. See below for more details


**Summary Of The Paper:**

The authors study phase transitions in linear regression with permuted labels for both the oracle setting (in which the regression coefficients are assumed to be known) and the non oracle setting (in which both the permutation and the regression vector are considered as unknown). The main result of the paper is an estimator for the signal to noise ratio above which recovery of the permutation matrix is possible in the absence of any estimate for the regression coefficients. The result relies on an estimator for the permutation matrix derived in a previous paper (Hang Zhang, Ping Li, Optimal estimator for unlabeled linear regression) and applies the machinery of Semerjian et al 2020 (cavity method) to derive the phase transition.




**Summary Of The Review:**

I don't think the paper can be accepted in its current state. Although the result is interesting, the paper should either be seriously reorganized or submitted to a journal which would allow for a longer and clearer exposition.  Detailed comments can be found below.


page 2
- The second item when you expose your technical contributions is not very clear

page 3

- In figure 1, it is difficult to read the expression in the red square

page 4

I’m in favor of removing the whole section as it appears exactly in this form in Mezard and Montanari Chapter 16. (A couple of comments can be found below)

- Section 3.2. When you say “here L and R are used to denote the positions of the nodes” do you mean the row vs column indexing ? This is unclear
- Your introduction of the messages in (3) is not clear. Honestly the \cong notation is misleading, why not use the traditional \leftarrow. What you are introducing here are the messages updates. Also you should really get rid of the \hat{m} e^{-\beta \pi_{i,k} E}… that appears in the second line and replace it with the RHS from the first update. Expanding the expression only makes things more confusing than they are
- Do you really need to keep the L and R indices, that makes the notations so much heavier
- I feel like the second equation between (3) and (4) in which you introduce the log ratio is redundant.
- Is \zeta (above (4)) the same as \zeta_{i,j} ? if so you should keep the same notation.
- The whole section on MP has to be rewritten. You introduce variables from nowhere. Below equation (4), what is \hat{\pi} ? We can somewhat guess  that it corresponds to the (estimated) permutation operator but for that same operator you used ‘\pi’ on page 2 and ‘pi^\sharp’ on page 3. Why not just keep pi for the entries in the matrix \Pi?
- It is very difficult to follow on your definition of the cavity fields in Equation (5). In fact you should rewrite the transition. At least add a reference to (5) in the paper of Semerjian. For someone who has not read the paper, it is absolutely unclear where the updated BP equations on the cavity fields come from (in particular the min which appears out of nowhere)
- Honestly, you should provide a clear reference to section (16) in Mezard and Montanari or (15) in Semerjian
page 5
- When you introduce the random variables H and \hat{H}, you skip a number of steps and that makes the reading of the paper quite painful. E.g. you again need to zero temperature limit for (7) to hold if I’m not wrong
- What you call the message flows are the cavity fields?
- You can introduce the independent copies because you have equality in the distributional sense (again I think you should specify this, otherwise you lose the reader who is not familiar with the cavity method)
- The equations in (7) are the same as (48a) and (48b) in Semerjian but if you look at (48b) the first update is carried only with probability (1-q). Also look at (29a) (29b)
- The transition from (4) to the condition H+H’>\Omega is unclear. Either provide a reference or expand
- The definitions of the random variables Omega and K are unclear. If I’m not wrong, you never introduce the K that appears in the statement of Theorem 1. If I’m not wrong the K has a precise definition as a lower bound for H following from a simplification of the Recursive Distributional Equations for H. This should be properly introduced.
- Also your notion of phase transition is unclear. What is meant by phase transition here? Once again, there is too much content and the essentials are missing. If you look at the Semerjian paper, those notion are carefully introduced. E.g. the authors take the time to explain that in the large n (or in your case t) limit, a value of K (and hence H since K lower bounds H) going to infinity implies that the only solution to the BP equations is the solution H = +infty. we are thus interested in the transition from the setting where there are possibly other solution (i.e. K\neq \infty) to this last setting. This explanation should appear in your paper
- In (12) you introduce an SNR but there is no explanation on where that SNR comes from or even what it encodes. We can sort of guess it is related to the sigma in (6) and on page 2 but the connection is unclear
- When you introduce the SNR, is that the SNR between Pi*X*B and W at which the transition occurs ? You should elaborate on this. Is the SNR related to sigma?

page 6
- Why is assumption 1 a good assumption ? You say “widely used assumption” but you don’t provide any reference

- Section 4.2. is particularly unclear.

- In the discussion before section 5, when you discuss subGaussian matrices What is ENER ?

- Honestly, your introduction of the two different approaches to derive the threshold SNR is unclear. From what I understand, you first use the solution of inf 1/\theta * \log(\sum_i \mathbb{E} e^{-\theta \Xi_i}) for which you have to compute the average  E{e^{-\theta \Xi}} exactly and then you use the taylor expansion approach (which you want to use in the non oracle case). You motivate the second approach by saying that it is not always possible to compute the expectation. Yet in this case (since you restrict to the identity) you just computed this expectation so why use the simplification? You should clearly explain why you discuss both approaches (i.e. add a short sentence such as “Solving the non oracle case requires an approximation on the expectation \mathbb{E} e^{\theta \Xi}. We first motivate such an approximation by showing that it can be related to the exact solution for the SNR, in the particular case of the identity”).

- From what I understand the conditions  inf_\theta (1/\theta) \log\left(\mathbb{E} Z * \mathbb{E}e^{-\theta\Xi}\right) and inf_\theta (1/\theta) \log(n \mathbb{E} e^{-\theta \Xi}) are the same. what you do is, in the first approach, you compute the exact expectation, in the other, you use an approximation. But this is very poorly explained
- Can you claim that your approximation is sound just because it works for the identity ?

page 7/8

- The connection between the oracle and the non oracle case is not very clear. In particular, why can you apply the same machinery in the non oracle case as the one you use in the oracle case? I.e. from the paper it seems straightforward that the phase transition in the non oracle setting occurs when -inf_theta \frac{1}{\theta} \log\left[\sum_i \mathbb{E} e^{-\theta\Xi_i}\right] or inf_\theta \log(\mathbb{E}Z \mathbb{E}e^{-\theta\Xi}). Yet this does not seem straightfoward to me. I.e. when you say “second, this estimator also exhibits a phase transition phenomenon, which behaves similarly to that in the oracle case”, can you expand a little more ?

- You should give a more detailed reference (indicate Algorithm 1?) to your 2020 paper (i.e. Zhang et Li) in which you introduce the estimator for Pi.

- How do you go from your estimator (i.e the 2020 one to the matrix Xi? further explanation is needed here). If I understand well, because your estimator does not rely on the regression coefficients, you dcan apply the BP equations to this estimator to study the phase transition. if so you should add that somewhere.

- In the statement of Theorem 2, you reference an equation (39) in appendix

- I would remove the proof details for Theorem 2 as it is just the computation of an expectation (i.e. everything can be done by means of Isserlis' theorem) and your explanation of the three phases is unclear. I think no one will blame you for not providing the details of the computation of an expectation. But if what you provide is unclear, then a the reader will start to question the whole paper.

- In the very last paragraph of page 8 you mention a gap between the theoretical value and the numerical value. What gap are you talking about? Where are the numerical experiments ? Are you talking about The result from table 3, then you should add a reference to that table.

page 9

- At the beginning of page 9 (first paragraph), you again indicate that numerical experiments confirm your results while not adding any reference to those experiments. Are you talking about Figure 2/Table 3 because the captions of those do not seem to relate to what you say on page 9
- I would consider removing section 6. The section should be expanded and is unclear in its current state. Why would you want to do partial recovery if you can apply the full recovery algorithm. You need to better motivate this (for example in terms of computation) or remove it.
- The discussion is not clear. From what I understand the SNR already represent the SNR at which the phase transition occurs. Then you indicate that there is a possible phase transition w.r.t tau ? what do you mean? there is a regime in which you can’t predict anything ? i.e. no SNR will enable the recovery? This also appears surprising to me that this regime corresponds to a very narrow subset of tau values (as your plot in Figure 2 seems to indicate). How do you explain this?

As a side note, I find it surprising that the Hamming distance of the permutation matrix to the identity seem to matter for the phase transition but maybe this is a well known fact.

================================================================

Typos/syntax errors

================================================================

page 1
- fisrt paragraph: “we have witnessed a revival of this problem due to its board spectrum” —> I guess you mean “broad”
- The result introduced in Zhang et Li 2020 is indeed very similar to the work of the paper
page 2
- When you say “the maximum allowed permuted rows”, do you mean the “maximum allowed number of permutations”?
- When you introduce your technical contributions, “By removing all function nodes except that corresponds to that index” —> “except the one that corresponds”
- Again, when you introduce your technical contributions : “an algorithm that converge in one step “ —> “that convergeS”
- “its performance almost match” —> “its performance almost matchES”

page 4

- Before Equation 5: “we can thus simply the MP update equation as” —> “we can thus simply write”?
- Before Equation 6: “where B^ sharp is given a prior” —> “B^\sharp is given as a prior” ?

page 6
- section 4.2. “to obtain a closed-formula” —> “A closed form expression”?
- First paragraph of section 4.2. “The computation of phase transition threshold is by setting”  —> “The phase transition can be estimated from“ ?
- In the title of section 4.2. : “APPROXIMATED COMPUTATION ” —> “Approximate computation”

page 7
- Computation of the mean: “easily we can verify” —> “we can easily verify”

page 8
- Phase 1 of the computation of the variance : “This decomposition is in the same spirit of the leave-one-out technique” —> “in the same spirit as …”
- Right before the subsection “Computation of Var \Xi”: “which consists the calculation of the following six terms” —> “which consists OF the calculation of…”
- Phase 2 of the computation of the variance : “we can exploit the independence among rows” —> “among the rows”
- Phase 2 of the computation of the variance: “which contains majority of terms” —> “which contains THE majority of THE terms”
- Same phase: “reduce the order of Gaussian random variables” —> “of THE Gaussian variables”
- Same phase: “by separately taking expectation’ —> “by taking THE expectation”
- When you explain phase 1 for the computation of the variance, “and leave the rest terms to matrix” —> “the rest OF the terms” or even better “the remaining terms”
- In the discussion paragraph at the end of the page: “we notice a increasing gap” —> “AN increasing gap“
- In the caption of table 3, “rate drops blow 0.05 ” —> “drops below”?

Page 9
- At the beginning of the first paragraph you say “Numerical experiments confirm our prediction ”. Which numerical experiments are you talking about ?
- In the conclusion: “For the oracle case where the signal B is given a prior ” —> “is given AS a prior”

---

> ### Author Response · Authors · 2022-11-19
> **Thank you for your review**
>
> Dear Reviewer: Thank you for the value insight and detailed comments. We understand that your main concern is the organization of the material. As you have summarized, to understand our work, readers need to be familiar with both 1) message-passing literature, techniques, and notation; 2) permutation recovery.   This is a high expectation because other message-passing and permutation recovery are difficult (and fairly new) research problems.  This perhaps also implies the novelty of our work.
>
> We appreciate your detailed comments on the notations and organization. We will certainly try our best to incorporate them into the revision. Again, since this paper involves two difficult topic: 1) message-passing; 2) permutation recovery, different readers will have different expectation on which materials are known and which are new (and should be placed in the main body (i.e., first 9 pages) of the paper). We understand your suggestion of moving materials on message-passing to the appendix, while we also anticipate that readers who are more familiar with permutation recovery would prefer to see introduction to message-passing in the main body of the paper. In any way case, it is our responsibility to find ways to satisfy the needs from both groups of readers, using the limited space. Thank you.
>
> We hope it is fair to say that other than the notation/organization issues, your reviews/comments suggest that our submission is a novel combination of both permutation recovery and message-passing. It is our understanding that conferences such as ICLR would like to see novel new research topics, and we hope our work fall into this category. Thanks again.
>
> We really like your "side note": *As a side note, I find it surprising that the Hamming distance of the permutation matrix to the identity seem to matter for the phase transition but maybe this is a well known fact.*   Our paper is the first time in the literature to theoretically demonstrate this interesting phenomenon.

---

### Author Response · Authors · 2022-12-10
**feedback on the rebuttal?**

Dear Reviewers and Area Chairs,

Thanks again for providing the feedbacks. We hope the code we sent has helped Reviewers better understand our algorithms and contributions.

As this is the very first work which involves two very different (and difficult to many) areas: (A) message-passing for phase transition, and (B) permutation recovery, we understand it might be quite difficult for readers to fully understand the contributions and the technical details. In particular, as the message-passing method itself is still under development (for example, for becoming fully rigorous), we have anticipated the challenges in publishing this work. Nevertheless, we appreciate the suggestions and comments. Please let us know if there are further questions which we could answer. Thank you.

The Authors.

---

### Decision · Program_Chairs · 2023-01-20

**Decision:**

Reject

**Justification For Why Not Higher Score:**

There are serious doubts about the generality of the proposed solution and clarity of the paper.

**Justification For Why Not Lower Score:**

N/A

**Metareview: Summary, Strengths And Weaknesses:**

The reviews overall point to important drawbacks. I think the reviews summarize well the strengths and weaknesses of the paper. In particular, the oracle case focuses on B being the identity and it is not clear whether the assumptions the theory based on belief propagation needs are still satisfied when the matrix B is something else. The readability and clarity of the paper is also an issue. Hopefully, the reviewer's feedback will help the authors to improve their work.